# A diagnostic interface for the ICOsahedral Non-hydrostatic (ICON) modelling framework based on the Modular Earth Submodel System (MESSy, 2.50)

Bastian Kern and Patrick Jöckel

Deutsches Zentrum für Luft- und Raumfahrt e.V. (DLR), Institut für Physik der Atmosphäre, Oberpfaffenhofen, Germany

*Correspondence to:* Bastian Kern (bastian.kern@dlr.de)

**Abstract.** Numerical climate and weather models have advanced to finer scales, accompanied by large amount of output data. The model systems hit the input and output (I/O) bottleneck of modern High Performance Computing (HPC) systems. We aim to apply diagnostic methods on-line during the model simulation instead of applying them as a post-processing step to written output data, to reduce the amount of I/O. To include diagnostic tools into the model system, we implemented a standardised, easy-to-use interface based on the Modular Earth Submodel System (MESSy) into the ICOsahedral Non-hydrostatic (ICON) modelling framework. The integration of the diagnostic interface into the model system is briefly described.

Furthermore, we present a prototype implementation of an advanced on-line diagnostic tool for the aggregation of model data onto a user-defined regular coarse grid. This diagnostic tool will be used to reduce the amount of model output in future simulations.

Performance tests of the interface and of two different diagnostic tools show, that the interface itself introduces no overhead in form of additional runtime to the model system. The diagnostic tools, however, have significant impact on the model system's runtime. This overhead strongly depends on the characteristics and implementation of the diagnostic tool. The diagnostic tool with high inter-process communication introduces large overhead, whereas the additional runtime of the diagnostic tool without inter-process communication is low. We briefly describe our efforts to reduce the additional runtime from the diagnostic tools, and present a brief analysis of memory consumption. Future work will focus on optimisation of the memory footprint and the I/O operations of the diagnostic interface.

## 1 Introduction

Resolution of numerical climate and weather simulations has advanced to finer scales, driven by increasingly potent high performance computing (HPC) systems. Throughput of high resolution simulations on modern multi-core systems increases, accompanied by large amounts of output data. These huge datasets pose challenges to the developer and the user, e.g. on how to cope with limited input and output (I/O) resources during data production, handling, processing, and archiving. Thus, the "workflow", i.e. how data produced by the model is handled, stored, and processed in subsequent steps, becomes more critical.

The performance of present day HPC systems reaches the Petascale, i.e. providing peak performances of more than $10^{15}$ operations per second (see http://top500.org/). While the increase in computational power follows still Moore's law, this does

not hold for the performance of the I/O (sub-)system. I/O operations are limited by the bandwidth the I/O system offers to the compute system. With increasing computational power, the ratio between I/O bandwidth and floating point operations per second decreases. Thus, for many applications the main bottleneck is I/O, potentially impacting the application performance on the next generation of supercomputers (Ali et al., 2009). Furthermore, storage capacity grows at a smaller rate as the computational speed (Kunkel et al., 2014; Kuhn et al., 2016). These problems especially persist for data intensive applications like numerical climate models, relying on fast data throughput and large storage capacity.

An obvious solution addressing the problems of data handling, the I/O performance gap, and the limited storage capacity, is to reduce the amount of output data. But a simple spatial and temporal reduction (coarse-graining) may prove counterproductive. When we apply post-processing tools to the coarsened datasets, it deteriorates advantages gained through finer resolved numerical simulations.

Recent studies on volume reduction of climate data focus on lossless and lossy compression of data on-line or as post-processing (Kuhn et al., 2016; Baker et al., 2016). We propose an approach of using diagnostic analyses, which are usually applied on the data after they are stored on disk, on-line during the model simulation. Depending on the analyses needed and selected, with the on-line diagnostic tools we might be able to drop parts of the output of variables on the full model resolution. This reduces data transfer over the computer network and disk storage occupation. The application of on-line diagnostic tools enables us to use the fully resolved data, to perform data reduction, and even to apply advanced diagnostics, which are not applicable off-line. On-line diagnostic tools can also be combined with any kind of data compression.

Such on-line diagnostics can be very simple and the implementation into the model system code can be straightforward. In most cases, however, the direct implementation of diagnostic tools in the model introduces dependencies throughout the code, which will be problematic with respect to maintenance of both, the model system and the diagnostic tool itself. This problem growth larger with the number of model variables accessed by the advanced diagnostic tool and the complexity of its code structure.

A direct implementation of diagnostics may also be difficult for users, who apply the model system but have low experience with the model code itself. We know from experience, that e.g. adding new variables to the model system or finding the right place to plug-in your diagnostic tool sometimes turns out to be cumbersome. Furthermore, unexperienced developers may not be aware of possible side-effects they introduce to the model system code and their impact on the model performance. A standardised, well documented, and easy-to-use interface supports users during the implementation of their diagnostic tools and helps to disentangle developments of the tools and the model system itself.

In this study, we present a first application of an interface for on-line diagnostic tools based on the Modular Earth Submodel System (MESSy, version 2.50, Jöckel et al. 2010) in the ICOsahedral Non-hydrostatic model framework (ICON, Zängl et al. 2015). In Sect. 2 we briefly describe the model system, the structure of the interface, and the implementation of the diagnostic interface into the model system. Then we present a prototype implementation of an advanced diagnostic tool for data aggregation on a user-defined regular coarse grid (Sect. 3). The performance of the diagnostic interface and two different on-line diagnostic tools is analysed in Sect. 4. In the end we present ongoing developments and future plans for applications (Sect. 5) and close with our conclusions (Sect. 6).

## 2 Model system

The work described here is part of the German-wide HD(CP)[2] (High Definition Clouds and Precipitation for advancing Climate Prediction, http://hdcp2.eu/) research initiative. HD(CP)[2] is funded by the German Federal Ministry of Education and Research (BMBF) and aims to improve our understanding of clouds and precipitation and their implication for climate prediction. The research attempt follows an integrated "three-pillar" approach combining model development and observations via a synthesis group.

HD(CP)[2] focuses on the development of a high-resolved model system and its application for simulations with a horizontal grid-spacing of about $100\,\mathrm{m}$ (Dipankar et al., 2015). To evaluate this model system on fine resolved scales, high-resolution observational datasets were obtained during the HD(CP)[2] Observational Prototype Experiment (HOPE) between April and May 2013 (see the ACP special issue: http://www.atmos-chem-phys.net/special_issue366.html). In the synthesis project, model and observational data are analysed and combined to evaluate and improve parameterisations in climate models.

One goal of the synthesis project is to develop advanced diagnostics and forward operators applicable to both, observations and model data. This approach of consistently processing data from observations and from model simulations enhances comparability between both. For this, we implemented a dedicated interface for the integration of enhanced diagnostic operators into the model system. We aim to provide a generalised and easy-to-use interface to plug-in existing and novel diagnostic tools for the application within ICON. The use of a dedicated diagnostic interface in ICON, furthermore, supports concurrent developments of the diagnostic tools and of the model system, which is still further enhanced during the project. A generalised interface also enhances the possibility of using the same diagnostic tools with other numerical models. Next, we will give a short overview of the involved components, before we present the implementation details of the diagnostic interface in the model system. Although, this study focuses on the implementation in ICON, the implementation in other numerical models would be similar.

### 2.1 ICON

The ICOsahedral Non-hydrostatic modelling framework (ICON, Zängl et al. 2015) is actively developed at the German Weather Service (DWD, Deutscher Wetterdienst) and the Max Planck Institute for Meteorology (MPI-M). The joint project targets a unified modelling system for application in global numerical weather prediction and climate modelling. Three main goals were defined during the initial phase, driving the implementation of the non-hydrostatic core of the model system (Zängl et al., 2015). These goals are

- the achievement of better conservation properties compared to the existing global model systems at the institutions, i.e. local mass conservation and mass-consistent transport, and the wish for energy conservation,

- the scalability of the model system on future parallel high-performance computing (HPC) platforms, and

- the availability of static mesh refinement, evolved during the model development into the capability of multiple one-way and two-way nested grids within one model application and an option for vertical nesting.

The model system is discretised on an unstructured triangular C-grid, derived from a spherical icosahedron using iterative refinement. Discretisation on this grid does not support exact global energy conservation, but the triangular grid was chosen because of the more convenient implementation of nesting using an hierarchical mesh refinement. Details on the discretisation of the equations of motion on the triangular C-grid and the numerical implementation of the non-hydrostatic dynamical core are described by Zängl et al. (2015).

## 2.2 MESSy

The Modular Earth Submodel System (MESSy, http://www.messy-interface.org/, Jöckel et al. 2005, 2010) is a framework for the implementation of (parts of) Earth System Models (ESMs). MESSy provides a standardised and generalised interface for the implementation and coupling of multiple ESM components (e.g. dynamical cores, physical parameterisations, chemistry packages, diagnostic tools, etc.) called "submodels". Currently, there are more than 60 submodels available, covering general infrastructure, diagnostics, atmospheric chemistry, and model physics. The MESSy interface was integrated into various numerical models (ECHAM5, Jöckel et al. 2005, 2010; COSMO, Kerkweg and Jöckel 2012a; CLaMS, Hoppe et al. 2014; CESM, Baumgaertner et al. 2016; see glossary for acronyms).

In geoscientific modelling, coupling of multi-institutional codes with generally different domain decompositions is a widely used approach for building model systems. In general, either *external* couplers or frameworks for *internal* coupling are applied. An extended classification of coupling methods can be found in Appendix A of Kerkweg and Jöckel (2012b) and in Jöckel (2012). An overview of different coupling techniques in Earth System Modelling is presented by Valcke et al. (2012). Common *external* couplers in the Earth System Model community are, e.g., OASIS3 (Valcke et al., 2006; Valcke, 2013), OASIS4 (Redler et al., 2010), and CPL6 (Craig et al., 2005), as used in the Community Climate System Model version 3 (CCSM3, Collins et al. 2006). Widely used examples for *internal* coupling are the Earth System Modeling Framework (ESMF, Collins et al. 2005) and the Community Climate System Model version 4 (CCSM4, Gent et al. 2011). This approach is also used in space weather modelling with the Space Weather Modeling Framework (SWMF, Tóth et al. 2005). Recently, Hanke et al. (2016) developed the C-library YAC (Yet Another Coupler), which provides parallelised and efficient algorithms for grid transformation, interpolation, and data exchange.

In contrast to the coupling of "domains", MESSy was originally developed to work on the same spatial domain and parallel domain decomposition as the basemodel, applying a formalised process based operator splitting (Jöckel et al., 2005). The original implementation was intended to equip the atmospheric general circulation model ECHAM5 (Roeckner et al., 2006) with additional processes for atmospheric chemistry (EMAC, ECHAM5/MESSy Atmospheric Chemistry model, Jöckel et al. 2006, 2010). Operator splitting as *internal* coupling method is implemented (implicitly and less formalised) in the numerical model codes anyway, to integrate the different processes. However, the operator splitting approach of MESSy proves more powerful, also allowing for coupling of different domains, e.g., demonstrated by the integration of an ocean subsystem (Pozzer et al., 2011). An extension by Kerkweg and Jöckel (2012b) allows for one-way coupling of different spatially nested domains using a server-client approach with point-to-point communication. Extension into a two-way nested atmospheric chemistry model system is currently under development.

The concept of the MESSy interface consists of a four layer structure for the integration of submodels. This ensures the disentanglement between infrastructure and submodels. The "basemodel layer" (BML) consists of a central clock and run-time control. In current implementations, the part of the basemodel is provided by a general circulation model, a numerical weather model, or a box model, extended by standardised calls to MESSy specific main entry routines. The "basemodel interface layer" (BMIL) provides the standardised entry points to plug-in submodels. This part is basemodel specific, as it provides the additional infrastructure of MESSy. The "submodel interface layer" (SMIL) connects the specific implementation of the process to the infrastructure in the BMIL. The specific implementation of the process resides in the "submodel core layer" (SMCL). In this layout, the BMIL and SMIL could be seen as layers, translating between the basemodel and the specific implementation of the submodel.

Standardisation of the connection between BMIL and SMIL ensures separation of developments in the basemodel and in submodels. Using MESSy in a new basemodel requires a one-time implementation of a basemodel-specific BMIL. Furthermore, for each new submodel a SMIL module has to be implemented. From the SMIL any legacy routines in the submodel core can be called, which opens the ability to reuse existing code. Porting submodels to different basemodels thus becomes straightforward. Only sometimes, this requires minimal generalisation in the SMIL, when the submodel was initially designed for a specific basemodel.

The user interface of the MESSy framework is based on Fortran95 namelists, offering control over infrastructure and submodels, including the possibility to (de)activate submodels and to change submodel specific parameters. This minimises the need for re-compilation of the model system when changing the combination of submodels and/or parameter values between different simulations.

## 2.3 Integration of MESSy into ICON

The implementation of the MESSy interface in ICON follows the MESSy standard. Calls in the ICON code to subroutines in the MESSy BMIL are enclosed in preprocessor directives (`#ifdef MESSY ... #endif`). *This allows a strict separation of ICON and MESSy code*, and guarantees the compilation of the ICON model without any missing dependencies, if the MESSy code is not included. Basemodel-specific code is avoided in the MESSy interface, but when necessary, it is also enclosed in preprocessor directives (`#ifdef ICON ... #endif`). To use the MESSy interface, the user just has to invoke the standard ICON configuration script with an optional parameter string (`./configure --with-messy`) and has to make sure, that the MESSy code is available in the model distribution under `externals/messy`.

We are still actively developing and extending the code, regularly merging developments from the main development branch of ICON and development branches of the HD(CP)[2] project. Due to the four layer structure, only sometimes minimal changes in the MESSy BMIL code are necessary to ensure a working model system including the diagnostic interface.

The output infrastructure of the diagnostic interface was extended to use the CDI (Climate Data Interface, https://code.zmaw.de/projects/cdi) library to provide data output consistent with the ICON basemodel. However, the interface does not support asynchronous data output. At the time being, the diagnostic interface provide only strict serial output to the diagnostic submodels. This is a very strong limitation, which will be removed soon, as a parallel version of CDI is being developed at

Deutsches Klimarechenzentrum (DKRZ, German Climate Computing Centre). As we will see in the performance tests, the limited output capabilities of the diagnostic interface have quite an impact on runtime. However, this impact is limited and we provide comparisons without any data output, to estimate the impact of the diagnostic calculations on the model system runtime.

The implementation as diagnostic interface in ICON is based on MESSy version 2.50, but does not include the whole MESSy software package. At the moment the infrastructure modules are implemented to some extend as needed (generic submodels BLATHER, CHANNEL, CONSTANTS, SWITCH & CONTROL, TIMER, TOOLS; cf. Jöckel et al. 2010). Additionally the TROPOP (TROPOPause diagnostics) submodel was adapted to support ICON as basemodel, in order to have a simple submodel for testing and demonstration. Extensions to the MESSy code of version 2.50 were necessary due to special requirements of the new basemodel. These changes include the implementation of the basemodel specific BMIL and major changes in the generic MESSy submodels CHANNEL and TIMER, which are described in Appendix A.

### 2.3.1 Hybrid parallelisation

Shared memory parallel programming is a common technique for introducing concurrency in computer code. This parallel programming model introduces threads working concurrently on the same memory (address) space. On most present-day HPC systems, shared memory access can be used only on a "per node" base. In such environments hybrid codes can be applied, using the shared memory programming model on the nodes and a distributed memory programming model across nodes. In many applications, this is a combination of OpenMP (Open Multi-Processing, http://openmp.org/) for intra-node parallelisation and MPI (Message Passing Interface, http://www.mpi-forum.org/) for inter-node exchange.

ICON is implemented to facilitate such a hybrid parallelisation. Thus, any extension should also use this combined programming model, to yield the most efficient use of computational resources. Inter-node parallelisation is achieved, using a parallel domain decomposition, so any submodel automatically benefits. Basically, it is easy to implement intra-node parallelisation, as OpenMP only requires specific directives in the code. However, it is not easy to say, where it is best to introduce this parallelisation in the submodels. Each submodel has its own computational aspects, and the gain in runtime through intra-node parallelisation critically depends on that. At this point we can give only a general advice to parallelise the submodel by introducing OpenMP directives to the loops, containing the major computational work. Each submodel developer should be aware of the hybrid parallelisation concept and should test their specific developments for an efficient implementation. It is planned to introduce OpenMP parallelisation in the BMIL in a future release, to benefit from automatic parallelisation of all diagnostic submodels. This, however, includes some necessary code refactoring and slight adaptation to the legacy submodels, which has not been started until now. In Sect. 4 we present some performance estimates for the inclusion of OpenMP parallelisation in two example submodels.

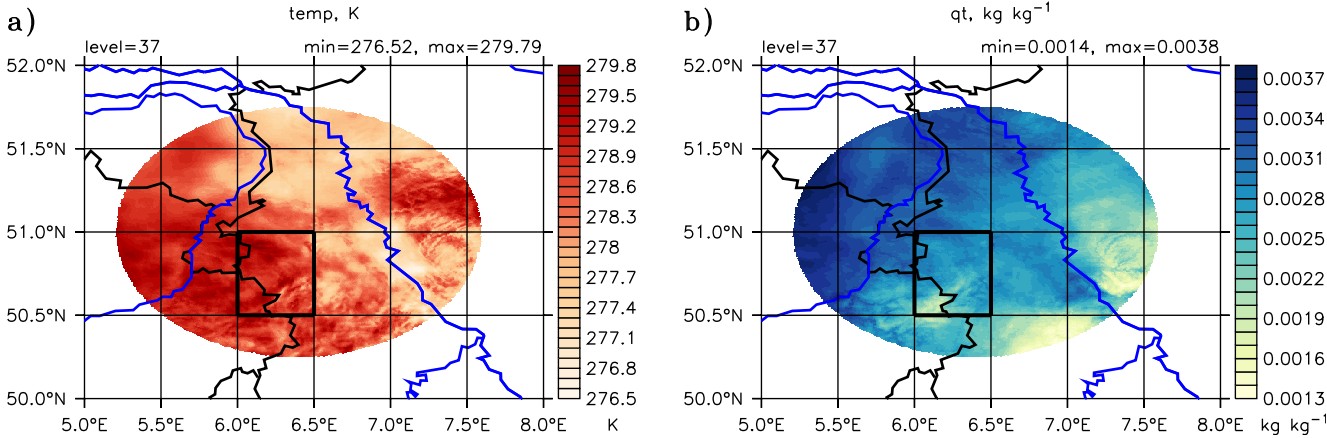

**Figure 1.** Snapshot at 24 April 2013, 01:00:00 UTC simulation time on model level 37 (approx. 1800 m) of the 3D ICON model output over parts of the Netherlands, Belgium, and western Germany. a) temperature, b) total water mixing ratio. Overlaid is the user-defined regular coarse grid ($0.5° \times 0.5°$) used to perform the on-line diagnostics in this study. For orientation note the country borders in black and larger rivers in blue.

## 3 GRAGG prototype

During the HD(CP)[2] project, we developed a prototype implementation of an on-line diagnostic tool, using the MESSy interface. The advanced on-line diagnostic tool GRAGG (GRid AGGregation) is capable to "aggregate" variables from the basemodel or submodels on a user-defined regular grid in different ways. User control is implemented according to the MESSy
5  standard via Fortran namelists.

At the time being on-line operations for the spatial average, the spatial sum, the discrete (binned) Probability Density Function (PDF), and the multivariate (for two variables) discrete (binned) jPDF (joint Probability Density Function) are implemented. Furthermore, an optional area-weighting can be selected by the user. For all options available via the Fortran namelist, see the GRAGG user manual in the electronic supplement, available online at doi:10.5194/gmd-0-1-2016-supplement.
10  Note that a temporal aggregation is not envisaged for GRAGG, because the generic CHANNEL submodel already allows time aggregation operations, such as minimum, maximum, average and standard deviation, etc. over the output time interval (see Section 2 and the Supplement of Jöckel et al., 2010). This facility can also be used for the variables calculated in GRAGG.

### 3.1 Implementation details of the prototype

The diagnostic submodel GRAGG operates in two phases, the initialisation phase and the time integration loop. During ini-
15  tialisation, the submodel processes the user namelist and initialises the memory. GRAGG stores, for each cell on the coarse user-defined regular grid, triangle indices from the native ICON grid, accessible in the physical memory of the respective processing entity (PE, corresponding to MPI task in this case). Simultaneously it counts the number of triangles present on the PE and their area in each coarse grid box. These values are communicated to the other PEs and combined to the total number of

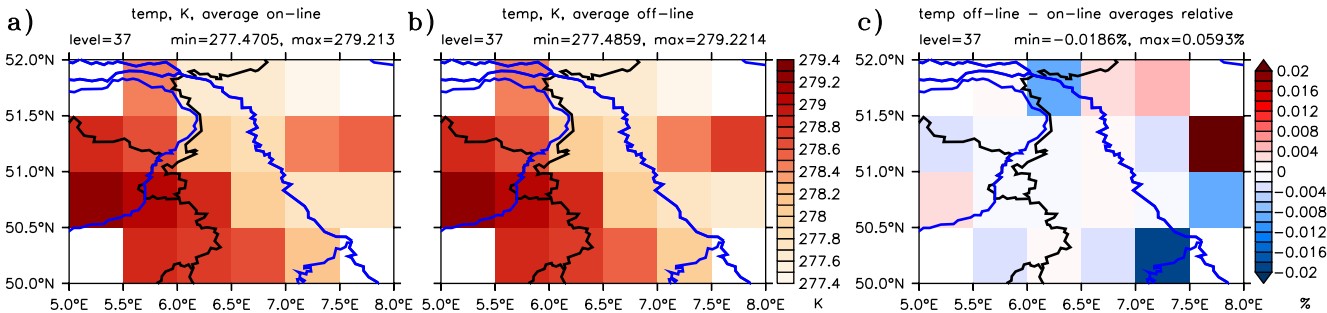

**Figure 2.** Temperature at 24 April 2013, 01:00:00 UTC simulation time on model level 37 (approx. 1800 m) averaged on the $0.5° \times 0.5°$ grid. a) output of on-line calculation from GRAGG, b) off-line calculation from 3D ICON model output, c) relative difference between off-line and on-line calculated values. For orientation note the country borders in black and larger rivers in blue.

triangles and their total area per grid box. Weighting coefficients are calculated for each triangle from its respective area and the total area in the coarse grid box.

During the time integration loop partial results are calculated on each PE. For summation and spatial averaging this is the sum over all values of the local triangles per grid box, with subsequent normalisation with the total number of triangles in each

box for the averaging operation. To achieve an efficient storage of jPDFs, GRAGG determines minimum and maximum of the distribution from the variables in every time-step per coarse grid box. This requires inter-task communication, as the memory locations of triangles contained in one box may be distributed over multiple MPI tasks.

For the (j)PDF, the bin width is determined from the user-defined number of bins and the calculated minimum and maximum in the coarse grid box. In a loop over all triangles in the grid box, GRAGG determines in which bin the value has to be put.

The number of values sorted into the bins are counted in a "bin vector", which holds the partial result of the (j)PDF for each grid box. After the loop over all triangles on the PE, the bin vectors for all grid boxes are normalised by division of the total number of triangles in the respective box. More details on the (j)PDF calculation can be found in the GRAGG user manual, which is part of the electronic supplement.

The last step takes place immediately before the output. Now the results from the partial calculation on the processes are

collected and summed up to form the overall result in the output fields. The advantage of splitting the calculation in determining partial results on the processes during the time loop and the final combination to a global result in the output time-step is the minimisation of inter-process communication during the time loop. However, we have to keep in mind, that during the time loop only partial results on the PEs are available.

## 3.2 Evaluation of GRAGG

To show the capabilities of the prototype GRAGG, we conducted a short ICON simulation with the diagnostic interface and the GRAGG submodel activated. Fig. 1 shows ICON model output snapshots of the innermost domain over parts of the Netherlands, Belgium, and western Germany at 24 April 2013, 01:00:00 UTC simulation time on model level 37 (approx.

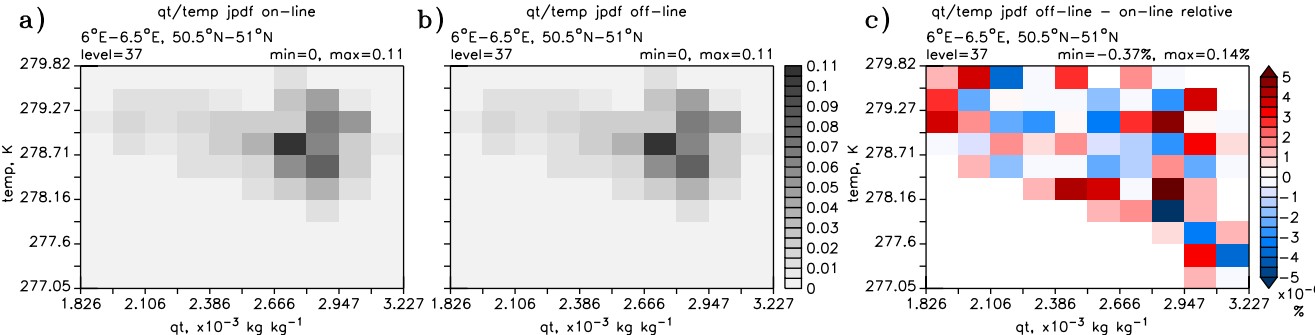

**Figure 3.** Joint PDF of water mixing ratio and temperature at 24 April 2013, 01:00:00 UTC simulation time on model level 37 (approx. 1800 m) in grid box $6.0°$ E - $6.5°$ E, $50.5°$ N - $51.0°$ N. a) output of on-line calculation from GRAGG, b) off-line calculation from 3D ICON model output, c) relative difference between off-line and on-line calculated values.

1800 m) for temperature and total water mixing ratio. The grid spacing of the innermost domain of the simulation is 312 m. GRAGG was set up to calculate the spatial average of temperature and the jPDF of total water mixing ratio and temperature on a user defined $0.5° \times 0.5°$ grid. We compare the results from GRAGG with off-line calculations based on the 3D ICON output, the Ferret scripts used to process the model data are available in the electronic supplement.

An example of the temperature averaged on the user-defined regular coarse grid on model level 37 (corresponding to approx. 1800 m) is shown in Fig. 2. The on-line calculated average temperature from the diagnostic tool is shown on the left (Fig. 2a), the same aggregation operation calculated from 3D ICON output off-line in the middle (Fig. 2b). The relative differences between the direct GRAGG output and the off-line calculation are lower than $6 \times 10^{-2}\%$ (Fig. 2c).

The jPDF of total water mixing ratio and temperature in the highlighted box of Fig. 1 on model level 37 is shown in Figs. 3a

and 3b for the on-line GRAGG output and the off-line calculation, respectively. As we see from Fig. 3, the relative differences mostly range between $-5 \times 10^{-6}\%$ and $5 \times 10^{-6}\%$, with one bin reaching $-0.37\%$ and one bin reaching $0.14\%$.

The discrepancies can be explained, as the off-line calculation operates on a single-precision output, whereas the on-line diagnostic makes use of the full double-precision values available during the model simulation. The examples shown here demonstrate the ability to get the same results from the on-line diagnostic as for the post-processing of model data.

**4   Application of on-line diagnostics**

In the following application, we use GRAGG to retrieve jPDFs (joint Probability Density Functions) of total water and temperature. As outlined in Sect. 3.1, GRAGG uses demanding communication patterns during the calculation of jPDFs. Thus, this task could be seen as one extreme with respect to the usage of computer resources. In this study we will use this submodel as example of a "communication bound" on-line diagnostic tool.

As an example for a submodel with low inter-task communication we will use VISOP (VIsual Satellite OPerator), developed at the Ludwig-Maximilians University (LMU) during the HD(CP)$^2$ project (Scheck et al., 2016). This submodel implements a

**Table 1.** Overview of the testcases for the performance tests.

| name | MESSy interface | submodels GRAGG | VISOP | optimisation OpenMP | reduced call frequency |
|------|------|------|------|------|------|
| ICON only | | | | 1 | |
| MESSy interface | X | | | | |
| MESSy + GRAGG | X | X | | | |
| MESSy + GRAGG OMP | X | X | | X | |
| MESSy + GRAGG reduced | X | X | | | X |
| MESSy + GRAGG red. + OMP | X | X | | X | X |
| MESSy + VISOP | X | | X | | |
| MESSy + VISOP OMP | X | | X | X | |
| MESSy + VISOP reduced | X | | X | | X |
| MESSy + VISOP red. + OMP | X | | X | X | X |

Open MP in the basemodel ICON is always enabled.

radiation calculation along model columns, i.e. stacks of triangles on the ICON grid. The implementation is column-based and inter-task communication only happens during output time-steps of the MESSy interface. This communication is necessary at the moment, because the interface performs only serial I/O (cf. Sect. 3.1). VISOP is an example of a "calculation bound" submodel, as most of the work in the time-step is spent in calculations over the model columns.

## 4.1 Model setup

We performed several runtime tests on Hochleistungsrechnersystem für die Erdsystemforschung 3 (HLRE-3, HPC system for Earth System research 3) at DKRZ to test the performance of the MESSy interface in ICON itself and the performance of the different submodels. We switched on or off specific components of the code to quantify the influence of the respective component on the model runtime. Table 1 lists all testcases and their specific set-ups. The tests consisted of simulations with the MESSy interface included and not included in the code, with the submodels GRAGG and VISOP switched on or off, with code modifications in the submodels to reduce the call frequency of calculations in the submodel to the output time-step only, and with OpenMP parallelisation of the loop(s) containing the main work of the submodel. A reduction of the call frequency can be only applied, if the calculations in the submodel do not rely on any data from other time-steps than the current one, and if calculated data are not required by other submodels.

For all tests we used revision r25413 from 16 November 2015 of the development repository `icon-diag-hdcp2-refactor` (https://svn.zmaw.de/svn/icon/branches/icon-diag-hdcp2-refactor). The code was compiled with Intel Fortran Version 14.0.3.174 Build 20140422 and linked against bullx MPI version 1.2.8.3 and netCDF 4.4.2 libraries. We configured the code with `./configure --with-fortran=intel --with-openmp --with-mpi=/opt/mpi/bullxmpi_mlx/1.2.8.3`

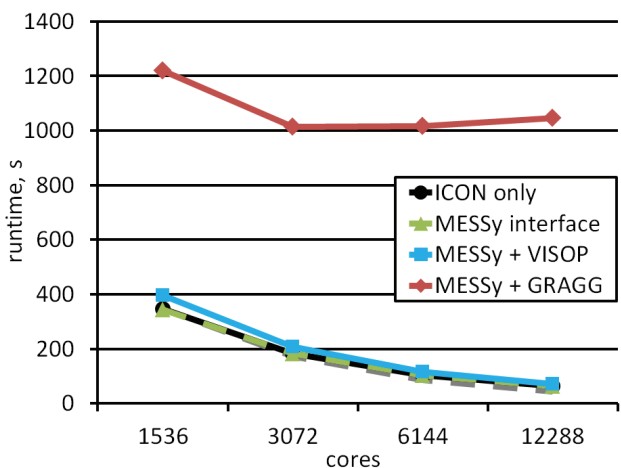

**Figure 4.** Runtime (s) for different testcases with deactivated output, measured with the internal timer "total" on 64, 128, 256, and 512 nodes (1536, 3072, 6144, and 12288 cores, respectively) of HLRE-3. The dashed gray line marks perfect scaling for "ICON only".

`--with-flags=hiopt`, for the inclusion of the MESSy interface we added `--with-messy`. This results in compiler flags (include and preprocessor flags dropped): `-openmp -O3 -mkl=sequential -pc64 -no-prec-div -no-prec-sqrt -fast-transcendentals -reentrancy threaded -xHost -vec-report1`.

The setup of all testcases was based on the experiment `exp.ICOLES_nestgrid`. It includes three nested domains over
Germany, with grid spacing of 1 249 m, 625 m, and 312 m and 302 912, 893 548, and 224 132 grid cells, respectively, resulting in 1 420 592 horizontal grid cells in total. Each nested domain has 50 vertical levels. The LES physics package with a two moment microphysics scheme is used (Dipankar et al., 2015). The simulations were integrated over 1 h simulation time, starting 24 April 2013, 00:00 UTC, with an integration time-step of 10 s. For all simulations we used 4 MPI tasks per node and 6 OpenMP threads per task. Asynchronous output of the ICON model was carried out by 6 processors.

When activated, the standard ICON output consisted of 35 meteogram stations with an output interval of 50 time-steps, the output of 13 model variables (2D) on the innermost domain every 15 min, and 20 cloud related diagnostic variables (2D) on all domains every 30 min. Output of variables related to the planetary boundary and the land model (32 2D and 2 3D variables, and 3 variables on 9 or 8 vertical levels) occurred every hour. Prognostic and radiation variables, and physical tendencies (altogether 24 3D and 27 2D variables) were written out every 3 h. The output of VISOP contained 10 2D variables, the output
of GRAGG as used in the testcases consisted of 6 variables on every model level on a reduced regular grid of $0.1° \times 0.1°$ (121 $\times$ 81 grid points), and one jPDF (3D on the reduced grid times 100 bins). In this study we used an output interval of 15 min for the diagnostic submodels.

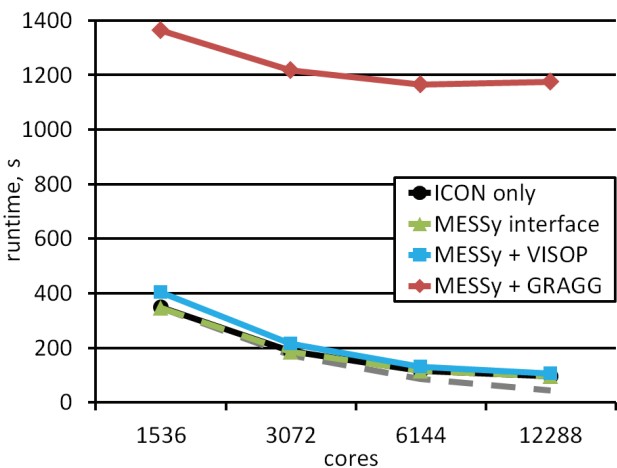

**Figure 5.** As Fig. 4 with output activated for ICON and the submodels.

## 4.2 Performance tests

For the analysis of model runtime, we utilised the timer debug mechanism already included in ICON. The results discussed here are the values reported for timer `total`. Tables with all runtimes can be found in the electronic supplement. In the following we use the term overhead $O_{\text{testcase,basecase}}(n)$ of a testcase relative to a basecase simulation, defined as:

$$O_{\text{testcase,basecase}}(n) = \frac{t_{\text{testcase}}(n)}{t_{\text{basecase}}(n)} - 1, \tag{1}$$

with $t_{\text{testcase}}(n)$ the runtime of a testcase on $n$ nodes, and $t_{\text{basecase}}(n)$ the runtime of a basecase on the same number of nodes. In this study "ICON only" with or without is considered as the respective basecase, in order to calculate the additional fraction of runtime caused by the diagnostic interface and the diagnostic tools.

The speedup $S_{\text{testcase}}(n)$ is calculated for each testcase simulation from the model runtimes $t_{\text{testcase}}$ on $n$ nodes relative to the corresponding simulation on 64 nodes:

$$S_{\text{testcase}}(n) = \frac{t_{\text{testcase}}(n=64)}{t_{\text{testcase}}(n)}. \tag{2}$$

Tables with calculated speedups for all testcases can be found in the electronic supplement.

The extension of ICON through the compilation of the model system with the MESSy interface does not introduce any significant overhead in terms of runtime of the model system (Figs. 4, 5). All increases in runtime due to the activation of the MESSy interface are below 0.1%. Measurable overhead is only introduced by activating diagnostic tools.

The scaling behaviour of the two diagnostic tools tested here with an increasing number of computer cores differ significantly. Whereas the overhead increases with higher number of cores for GRAGG, it decreases for VISOP. In both cases this behaviour is very similar regardless of activated output. From these results we can see the different nature of the two submodels. Whereas GRAGG is bound by inter-task communication, which increases the overhead with additional tasks, VISOP is bound by its calculations, which results in a better scaling with additional tasks (at least up to the number of 12288 cores, tested here).

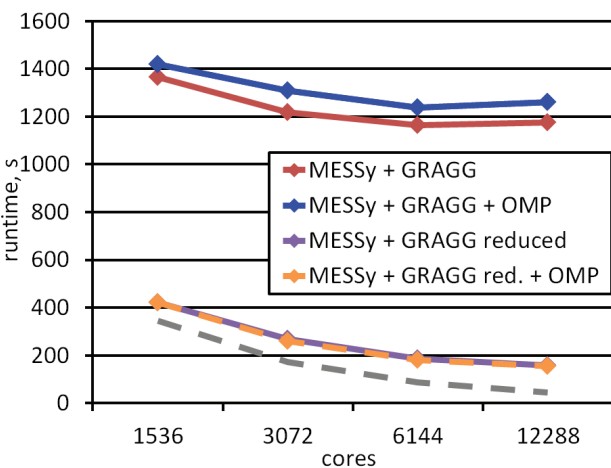

**Figure 6.** Comparison of the runtime (s) of GRAGG and versions of GRAGG with reduced call frequency, OpenMP, and a combination of both. Output was activated in all testcases.

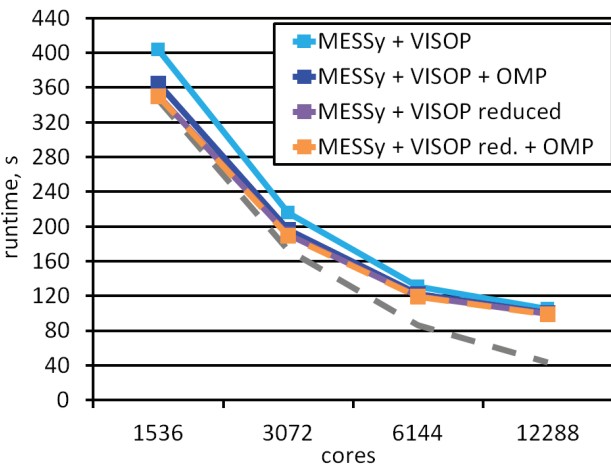

**Figure 7.** Comparison of the runtime (s) of VISOP and versions of VISOP with reduced call frequency, OpenMP, and a combination of both. Output was activated in all testcases.

To optimise the runtime behaviour of the two submodels we explored the possibility of making efficient use of the hybrid parallelisation approach by introducing OpenMP regions. Furthermore, to reduce the overall runtime of the model, we studied the effect of a reduced call frequency of the submodels. For this, we reduced the call frequency of the submodels to the output time-step of the corresponding submodel (15 min).

5      The comparison of the different optimisations for GRAGG and VISOP are shown in Fig. 6 and Fig. 7, respectively. In Fig. 8 the comparison of both testcases with the lowest runtime for the submodels GRAGG and VISOP, respectively, and the "ICON only" basecase is shown.

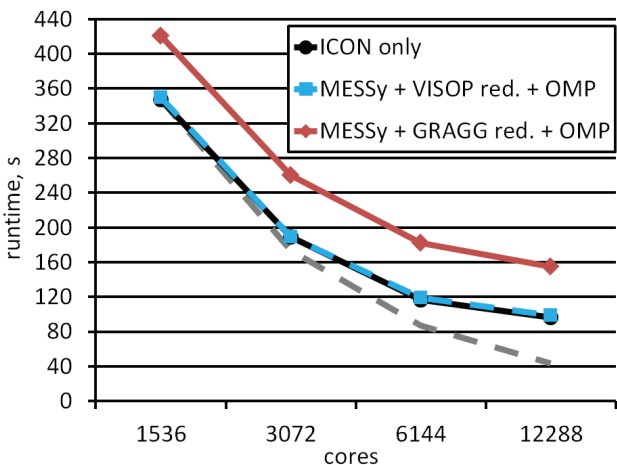

**Figure 8.** Comparison of the runtime (s) of "ICON only", and the testcases with the lowest runtime for activated GRAGG and VISOP. Output was activated in all testcases. The dashed gray line marks perfect scaling for "ICON only" with activated output.

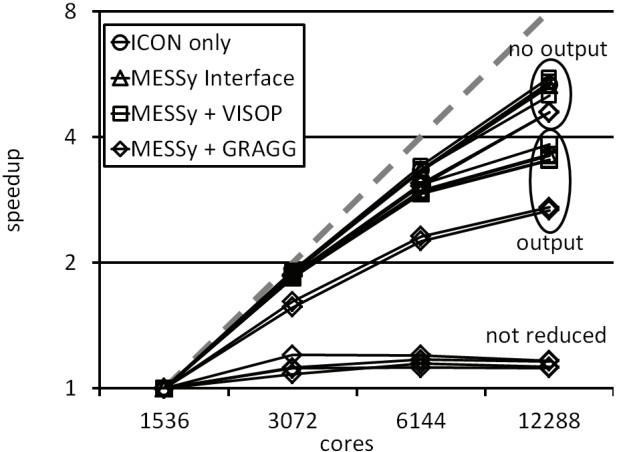

**Figure 9.** Speedup calculated to the baseline of 64 nodes (1536 cores) for each testcase. Symbols match those in Figs. 4–7. Labels mark simulations with and without output. For "MESSy + GRAGG", the lower four series are speedups for simulations with GRAGG operating on the full temporal resolution. The dashed gray line represents optimal speedup.

As the bulk in runtime of GRAGG results from MPI communication tasks, GRAGG does not benefit from OpenMP parallelisation of its calculation loops (Fig. 6). The total runtime even increases, due to the additional but inevitable overhead introduced by OpenMP. The additional MPI communication for the determination of minimum and maximum causes bulk of the additional runtime (cf. Sect. 3.1). The only applicable strategy to reduce the runtime of GRAGG is to restrict calls of the submodel to output time-steps only. This reduces the overhead from GRAGG compared to "ICON only" (no output) to 2.6% and 19.37% for 64 and 512 nodes, respectively. Between the versions of GRAGG with and without OpenMP parallelisation

only slight differences in the overhead exist. In the case we activate the output, overheads for this optimised version of GRAGG increase to 20.86% and 60.79%, respectively. The reduction of the call frequency of GRAGG yields a better scalability of the model system, resulting in a higher speedup (Fig. 9). For GRAGG with reduced call frequency and output, the speedup on 512 nodes is 2.71 and 2.67, with and without OpenMP, respectively. The values for the testcase without output are 4.58 and 4.59,

respectively. GRAGG operated on the full temporal resolution only reaches speedup values slightly more than 1 (Fig. 9). This shows a bad scaling behaviour for this MPI communication intensive tool.

For VISOP we see the benefit of parallelising the submodel using OpenMP (Fig. 7). This significantly reduces the overhead for VISOP by introducing simple OpenMP directives for the main calculation loop. Without OpenMP, the lowest overhead from VISOP is reached for 512 nodes, achieving values of 9.13% and 10.6% with and without output, respectively. Overall,

with a reduced call frequency and OpenMP parallelisation we can reduce the overhead of VISOP to 2.72% (for 512 nodes and activated output) and lower. Thus, with the optimised version of VISOP the total model runtime of 99.08 s (without initialisation) is only increased by 2.67 s.

The speedup for VISOP on 512 nodes is more than 5 without output and reaches more than 3.5 with output in all cases (Fig. 9). These values are above the speedups for the testcases with GRAGG. A reduced scalability because of the limited

I/O capabilities of the diagnostic interface (cf. Sect. 2.3) is obvious. But all speedups are comparable to the values obtained from the testcases of the ICON model without the diagnostic interface. The worse scaling for GRAGG compared to VISOP in all testcases with diagnostic output is caused by the larger amount of data written out and the additional MPI communication during each output step. The bad scaling behaviour of GRAGG operating on the full temporal resolution is caused by the intensive inter-task communication during each time-step (cf. Sect. 3.1).

The initialisation of the submodels takes only a few percent of the time the total model system needs for initialisation. We do not present any absolute values here, as there is a large variability in the times measured for initialisation. The initialisation includes extensive I/O operations for reading in the model grid and the domain decomposition, and hence the time for initialisation is influenced by I/O system usage of other applications on the HPC system.

With all configurations we reach an acceptable speedup, which obviously is reduced in testcases with activated output (Fig.

9). This is related to the I/O bottleneck and how I/O operations are currently implemented in the diagnostic tools. Whereas ICON uses an asynchronous output strategy, which enables execution of I/O (partly) in parallel to the model calculations, the output from submodels using the MESSy infrastructure is at the moment strictly serial. The runtime of the testcases with optimised code of VISOP and GRAGG including output are shown in Fig. 8. We find a very low overhead for VISOP. The additional runtime for GRAGG ranges between 58.6 s and 72.56 s for the fully optimised testcase with output. Compared to the

simulations with the optimised GRAGG without output, 79.8% to 89.5% of the additional runtime (512 nodes and 64 nodes, respectively) is caused by the activation of output. To increase the speedup and reduce the overall time to solution, efficient and parallelised output is a critical topic we are working on.

For the MESSy interface and the diagnostic tool VISOP, the memory consumption of the extension compared to the ICON model itself is modest (Fig. 10). Generally, the memory consumption per node is larger for the testcases with output activated.

Simulations with GRAGG have larger memory demands compared to the simulations with VISOP or without any submodel.

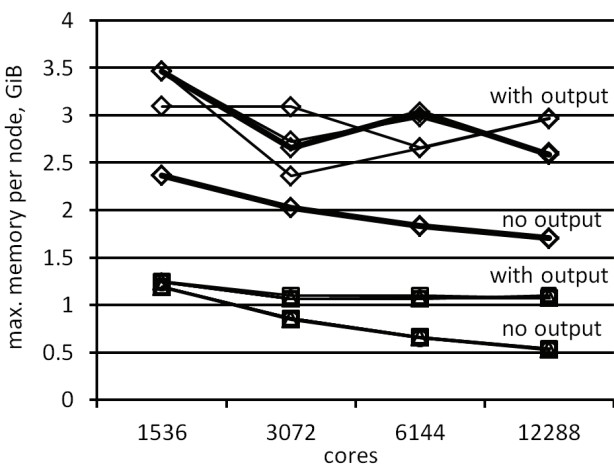

**Figure 10.** Maximum of memory consumption per node. Symbols as in Fig. 9.

This is due to the higher buffering demands of the increased MPI communication and internally used (temporal) fields. The largest memory consumption is reached for the simulations with GRAGG and activated output.

In general the memory demands differ substantially, depending on how much internal and output variables the submodels need. We are further investigating the memory demands of the model system. We want to reach a memory footprint as small as possible, especially facing the actually decreasing ratio of memory per compute core on HPC systems. This should also be kept in mind, when developing novel diagnostic tools to be run on-line in the model system.

## 5   Ongoing and future developments

At the moment we are porting our developments of the diagnostic interface and the diagnostic tools to the Jülich Blue Gene/Q HPC system JUQUEEN. First tests on this IBM PowerPC architecture are running successfully, but at the moment these tests are constrained to a fixed combination of 4 MPI tasks per node in combination with 4 OpenMP threads per task. The biggest challenge on that system is the low amount of available memory per core of 1 GiB, so we have to optimise our implementation towards a minimum memory footprint.

Furthermore, more diagnostic submodels are being developed during HD(CP)$^2$ in the areas of satellite simulators (COSP, Cloud Feedback Model Intercomparison Project's Observation Simulator Package, Bodas-Salcedo et al. 2008) and on-line trajectories. Our future plans include the implementation of an on-line feature identification and tracking system, and further forward operators, for improved comparison between model simulations and measurements.

We are working on further optimisation of the diagnostic interface and tools together with scientists of DKRZ in the second phase of HD(CP)$^2$. This includes the integration of parallel output in the interface, as soon as the improved parallel version of CDI is available in ICON.

Last, but not least, as there is already a re-gridding tool available in the MESSy framework, an integration into the GRID submodel (Kerkweg and Jöckel, 2015) with further generalisation of the aggregation developed in this study seems beneficial.

## 6 Conclusions

We presented the implementation of parts of the Modular Earth Submodel System (MESSy) into the ICOsahedral Non-hydrostatic (ICON) model framework for the application as diagnostic interface. This defines a generalised and easy-to-use interface for the implementation of on-line diagnostic tools. Because of the generalisation, the inclusion of the interface into other numerical models provides the possibility to operate the same diagnostic tools. This enhances the consistency of output between different models and reduces the need for re-implementation of already existing and working code.

We will be able to reduce I/O operations during future model simulations, as we write out on-line aggregated variables and reduce the need of storing high resolution 3D fields of model variables. Furthermore, as we reduce the need for (off-line) post-processing to derive diagnostic variables, we can reduce further sub-sequent access to the I/O system. However, the use of on-line diagnostic tools during the model simulations increases the runtime of the simulation, but we have to keep in mind, that post-processing tools do not come without computational costs.

There is a trade-off between the time needed for additional computations and the time needed for access to the I/O system. On the one hand, we could stay with the computational demands of the model system, the post-processing tools and their access to the I/O system. On the other hand, we could increase computational demands for a simulation, while reducing the demand for post-processing and the overall access to the I/O system. When the simulation is highly data-intensive, which is the case for high resolution simulations of ESMs, there is a chance to reduce the "time-to solution" on systems with an I/O bottleneck. The increase in runtime of the model simulation due to additional on-line diagnostics can be compensated by the omission of post-processing and by lower time spent for I/O operations.

We showed that the integration of the diagnostic interface itself has no impact on the model system runtime and memory consumption. The overhead introduced by diagnostic submodels, however, can be substantial, but could be reduced by thorough code optimisation. We reached a maximum overhead of 60.79% for the "communication bound" example of the jPDF diagnostic in GRAGG and a maximum overhead of 2.72% for the "computational bound" example of the satellite operator VISOP. Both simulations applied a reduced call frequency of the diagnostic tools and included output. Our future work will focus on the optimisation of the remaining I/O operations and the optimisation of the memory footprint. Furthermore, we will support the development efforts of additional on-line diagnostics utilising the MESSy interface.

The ratio between I/O bandwidth and peak performance of high performance systems decreased in the recent years, and probably will continue to do so for future systems, as the performance gap widens. Additionally the increase in storage capacity has slowed down recently. We have to focus on these problems during the design and provision phase of future HPC systems. Apparently, there will be no quick solutions to the problems, as the widening of the performance gap might persist for decades. For now, we will have to focus on software solutions to bridge this gap by providing efficient I/O operations, making use of parallelisation and appropriate I/O middleware in our codes, and also by reducing the amount of output data. Recent studies

focus on data compression as a (transparent) post-processing step, either when the data is stored, or even on-line before data is transferred over the computer network to increase communication performance.

We suggest the application of on-line diagnostic tools to reduce the volume of data from numerical simulations. This will require a largely modified workflow for scientists, as appropriate analyses have to be selected and on-line tools have to be developed before the simulation starts.

Checkpointing is already applied by model systems to overcome restrictions in compute times of job schedulers, and to restore simulations in case of software or hardware failures. Even when cutting the "full" model output to a minimum, a subsequent re-calculation with increased output volume and frequency is therefore possible. Thus, scientists should not be afraid of "losing" data, as a combination of all methods should suit their needs.

Currently applied workflows may not retain the same on future HPC systems. A certain degree of development and optimisation of current code and the application of novel methods are required. Here, a generalised interface for on-line diagnostic tools is extremely useful.

**Code availability**

The Modular Earth Submodel System (MESSy) is continuously further developed and applied by a consortium of institutions. The usage of MESSy and access to the source code is licensed to all affiliates of institutions which are members of the MESSy Consortium. Institutions can be a member of the MESSy Consortium by signing the MESSy Memorandum of Understanding. More information can be found on the MESSy Consortium website (http://www.messy-interface.org). The MESSy infrastructure submodels are freely available from the authors.

ICON (ICON atmosphere) will be made available under the ICON Software License Agreement ISLA version2.1, which will be a common SLA of the German Weather Service DWD and the MPI-M. Additional details for licensing ICON can be found at http://www.mpimet.mpg.de/en/science/models/license/.

For model developers, who obtained the proper licenses, a development version of the code is available from the authentication restricted repository at https://svn.zmaw.de/svn/icon/branches/icon-diag-interface.

**Appendix A:  MESSy specific code extensions**

The generic MESSy submodel CHANNEL (Jöckel et al., 2010) implements an interface for exchange of data between the basemodel and submodels, and among submodels, and handles the export of data to files. It is implemented in an object-oriented way and offers extended control of data flow, e.g. an enhanced user defined output control.

Time management is controlled via the generic submodel TIMER, which is based on the timer of the atmospheric general circulation model (GCM) ECHAM5. For the re-implementation as generic MESSy submodel, enabling its application for various basemodels, the timer relevant code was extracted from the GCM, keeping its original functionality and namelist syntax (Jöckel et al., 2010).

In the following sections, we present extensions to the development cycle 2 of the MESSy interface and its generic submodels CHANNEL and TIMER (Jöckel et al., 2010), resulting from special requirements of ICON as basemodel. More details on the MESSy interface and the implementation of submodels can be found in Jöckel et al. (2005, 2010). An extended version of the user manual for CHANNEL is included in the electronic supplement of the present paper.

## 5    A1    CHANNEL

The generic MESSy submodel CHANNEL manages data objects in a hierarchical structure build up from *channels* and *channel objects*. Details can be found in Jöckel et al. (2010) and the CHANNEL user manual available from the electronic supplement.

ICON provides the capability to be operated with refined nests, called *domains* or *patches*. One-way or two-way nesting can be used in the global configuration of the model as well as in the limited area mode. When a MESSy BMIL subroutine is called, the *patch* on which the basemodel is operating at that moment is passed to the subroutine by a parameter `patch_id`. The information is available to MESSy submodels via the variable `current_patch` (see below).

Although nesting of refined *domains* is supported in MESSy (Kerkweg and Jöckel, 2012b), this is only available for *direct external coupling* (cf. Appendix A of Kerkweg and Jöckel, 2012b), which utilises different program units for the different nests. In this configuration CHANNEL has not to be aware of different nests, as the nested *domains* reside in different namespaces, and are separated physically in memory and storage. As refined nested *patches* in ICON use internal direct coupling in one executable, we had to extend CHANNEL to support multiple *patches* for data management and storage. Changes in the application programming interface (API) are implemented using optional subroutine parameters, ensuring full backward compatibility for existing submodels. Here, we describe the structural changes of the CHANNEL source code and the corresponding changes in the API of CHANNEL subroutines in brief.

The complete list of *channels* is stored in a concatenated list. Until now, one pointer `GCHANNELLIST` is defined in `smcl/messy_main_channel.f90` pointing at the first element of the *channel list*. To support more than one *patch*, we introduced a wrapper `t_channel_list_ptr` for the *channel list*, which allows for allocating as many `channel_list_ptr` as there are *patches* in a simulation. In the standard case, when the basemodel does not allow native nesting (or no nests are defined), only one `channel_list_ptr` is allocated. This allows for backward compatibility by applying all operations on *channels* and *channel objects* only to the elements of the first (and only) *channel list* in that case.

To avoid the need of explicitly passing the *patch* on which the basemodel is currently operating to every subroutine, a public variable `current_patch` was introduced to the CHANNEL source code (in `smcl/messy_main_channel.f90`). This variable is set in the BMIL and can be accessed by any submodel via the Fortran `USE`. In CHANNEL subroutines the optional parameter `patch_id` controls which *patch* is accessed. Access of *channels* and *channel objects* from subroutines unaware of multiple *patches* is transparent, as in calls without the `patch_id` set, the subroutine works on the first *patch*. This allows for full backward compatibility of legacy submodels designed to work with basemodels without support of native nesting.

The concept of *representations* and *dimensions* as basic entities of the CHANNEL submodel is still valid in the implementation for ICON. As sizes of *dimensions* in different *patches* usually differ, these entities are stored separately for each *patch*. This concept allows to use identical names on the different *patches*, while additionally providing the `patch_id` when access-

ing these entities. This is transparent for the user of CHANNEL, but the submodel developer has to take care, if the submodel uses *representation* or *dimension* ID-numbers directly. In this case, the `patch_id` has to be used as index to the list of the *representation* or *dimension* ID-numbers.

## A2 TIMER

Until now, MESSy uses the generic TIMER submodel, re-implemented from the extracted time management code of the GCM ECHAM5 (Roeckner et al., 2006). It uses seconds as base unit and is synchronised to the basemodel timer each time-step. For high resolution simulations using a grid spacing in the order of $100\,\mathrm{m}$ – as it is planned for the HD(CP)$^2$ simulations – a smaller granularity in time management is needed. Our extension provides a granularity of milliseconds. Future plans include the change of the MESSy time management to an external library (MTIME), to be consistent with the ICON time management and to make use of its enhanced capabilities.

## A3 Submodels

All extensions to MESSy are implemented in a way to ensure backward compatibility. Thus, only minor updates to the SMIL modules of existing submodels are necessary.

The major modification was introduced, because we need to provide the ability of submodels to work on all patches defined by the basemodel. For this, a standardised subroutine `<submodel>_set_pointer` is called every time the simulation continues on a different patch. The purpose of this subroutine is to set the local pointers of the submodel to the appropriate memory in the current *patch* of the basemodel.

## Appendix B: Glossary

**API**  Application programming interface

**basemodel**  a general circulation model, numerical weather model, or box model extended by calls to *MESSy* specific routines

**BMBF**  German Federal Ministry of Education and Research (Bundesministerium für Bildung und Forschung)

**BMIL**  Basemodel interface layer

**BML**  Basemodel layer

**channel**  a set of *channel objects* and additional meta information in the CHANNEL *submodel*

**channel list**  in this structure, all *channels* are stored

**channel object**  data and its meta information

**CLaMS**  Chemical Lagrangian Model of the Stratosphere

**CDI**  Climate Data Interface

**CCSM**  Community Climate System Model

**CESM**  Community Earth System Model

**COSMO**  Consortium for Small-scale Modelling model

**COSP**  Cloud Feedback Model Intercomparison Project's Observation Simulator Package

**dimension**  describes the structure of data in a particular spatial or temporal direction

**DKRZ**  German Climate Computing Centre (Deutsches Klimarechenzentrum)

**domain**  spatial extend on which the model operates, different *domains* can be nested

**DWD**  German Weather Service (Deutscher Wetterdienst)

**ECHAM5**  European Centre HAMburg general circulation model, version 5

**EMAC**  ECHAM5/MESSy Atmospheric Chemistry model

**ESM**  Earth System Model

**ESMF**  Earth System Modeling Framework

**GCM**  General circulation model

**GRAGG**  GRid AGGregation submodel

**HD(CP)$^2$**  High Definition Clouds and Precipitation for advancing Climate Prediction

**HLRE-3**  HPC system for Earth System research 3 (Hochleistungsrechnersystem für die Erdsystemforschung 3)

**HPC**  High Performance Computing

**ICON**  ICOsahedral Non-hydrostatic model system

**I/O**  Input and Output

**jPDF**  joint Probability Density Function

**LMU**  Ludwig-Maximilians University, Munich

**MESSy**  Modular Earth Submodel System

**MPI-M**  Max Planck Institute for Meteorology

**MPI**  Message Passing Interface

**OpenMP**  Open Multi-Processing

**patch**  spatial extend on which the model operates, different *patches* can be nested

**PE**  Processing entity

**representation**  describes the underlying geometric structure of data, combination of *dimensions*

**SMCL**  Submodel core layer

**SMIL**  Submodel interface layer

**submodel**  a tool or extension, which is plugged in via the *MESSy* interface

**SWMF**  Space Weather Modeling Framework

**VISOP**  VIsual Satellite OPerator submodel

**YAC**  Yet Another Coupler

*Acknowledgements.* This work has been funded by the German Federal Ministry of Education and Research (BMBF) in the HD(CP)[2] (High Definition Clouds and Precipitation for advancing Climate Prediction) initiative as Research for Sustainable Development (FONA, http://www.fona.de/) under grant number 01LK1213C. Parts of this work have been supported by the European Union via funding of the
project IS-ENES2 (Grant Agreement Number 312979).

We thank P. Adamidis, A. Dipankar, T. Jahns, M. Hanke, the staff at DKRZ and the ICON developers for their support. We thank C. Meyer and L. Hoffmann for their support during the porting of our developments to JUQUEEN. We are grateful to L. Scheck for the provision of his code of VISOP and to J. Quaas for the provision of his jPDF code and the support during the development of GRAGG. We thank A. Pfeiffer for the fruitful discussion and the internal review, and A. Kerkweg for the comments on our manuscript.

The authors gratefully acknowledge the Deutsches Klimarechenzentrum (DKRZ) for providing computational resources on the HLRE-2 supercomputer, the HLRE-3 migration system, and the HLRE-3 supercomputer, as well as for technical support during the project.

The authors gratefully acknowledge the Gauss Centre for Supercomputing (GCS) for providing computing time on the GCS supercomputer SuperMUC at Leibniz Supercomputing Centre (LRZ, www.lrz.de) and through the John von Neumann Institute for Computing (NIC) on the GCS share of the supercomputer JUQUEEN at Jülich Supercomputing Centre (JSC). GCS is the alliance of the three national supercomputing
centres HLRS (Universität Stuttgart), JSC (Forschungszentrum Jülich), and LRZ (Bayerische Akademie der Wissenschaften), funded by the German Federal Ministry of Education and Research (BMBF) and the German State Ministries for Research of Baden-Württemberg (MWK), Bayern (StMWFK) and Nordrhein-Westfalen (MIWF).

The authors wish to acknowledge use of the Ferret program for analysis and graphics in this paper. Ferret is a product of NOAA's Pacific Marine Environmental Laboratory, information is available at http://ferret.pmel.noaa.gov/Ferret/.
We thank Ilja J. Honkonen and one anonymous referee for their comments and valuable input to our manuscript.

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
