# Peer review of "A diagnostic interface for the ICOsahedral Non-hydrostatic (ICON) modelling framework based on the Modular Earth Submodel System (MESSy, 2.50)"

_Geoscientific Model Development, 2016_

## Referee Comment (RC1) · I. Honkonen (Referee) · 2 Aug 2016

Overall the paper is quite good but a some clarifications would allow the presented work to be compared with existing literature in space sciences where similar work already exist. My answers to GMD review criteria are at the end.

The authors implement a module for ICON modeling framework that allows run-time post-processing (PP), for which they use the term online diagnostics, of simulation results. Supported PP operations are the calculation of spatial average, spatial sum, discrete probability density function (PDF) and a two variable joint PDF. These can be

calculated over a rectangular (in latitude and longitude) grid defined by the user. In a parallel run, if a cell of the PP grid spans more than one simulation cell, data required by PP operations is communicated between all processes that have simulation cells within the PP cell.

The term post-processing seems applicable here as the diagnostic module does not affect simulation results. The same term is also used e.g. in OpenFOAM documentation which shows an example of streamlines calculated at run-time (http://www.openfoam.com/version-v3.0+/post-processing.php retrieved on 2016-08-01). On the other hand the presented module could also be described as a one-way coupled model configurable with Fortran namelists.

Overall it seems that the presented implementation can be divided into two distinct parts: 1) A new one-way coupled model for the MESSy framework that allows new variable(s) to be added to the framework relatively easily and allows calculations performed by the model to be chosen at the start of simulation. 2) A method for moving data from simulation grid to the PP module grid in a transparent way that supports different grid geometries, cell sizes and aggregation of all data from simulation cells into overlapping PP cells.

Major comments

Item 1) above looks like a regular MESSy component which uses the same grid as the basemodel, would that be a fair statement? Can several PP modules run simultaneously on different sets of processes? Can a PP module use data from other PP modules or only simulation results i.e. non-PP or basemodel data? Is two-way coupling of PP modules possible, i.e. can two PP modules use each others' output data as their own input?

Item 2) above allows a PP module to use a different grid from the one used by basemodel. In the presented work, cells of PP grid are larger than in simulation grid, is it possible to make PP grid cells smaller than in simulation grid?

Taken together the functionality of items 1) and 2) are quite similar to e.g. the Space Weather Modeling Framework (SWMF, doi:10.1029/2005JA011126) which handles two-way data exchange between separate models running in parallel on different grids, although SWMF seems to lack the ability to collect all data from multiple smaller cells of the source grid into one cell of the destination grid. It would be helpful to clarify how the functionality presented here relates to that available in SWMF.

The support, or whether there is any, for temporal analysis within the PP module was unclear to me. Is it possible to process simulation data over several time steps while calculating one diagnostic output, e.g. can the minimum and maximum values of a variable over a certain time range in each cell of the PP grid be determined? This would determine the min and max values more reliably than calculating them from saved model output which presumably would not be feasible to do for every simulation time step.

If the above is possible, then are MPI operations within each PP grid cell available at each simulation time step, or only at the end of a PP step? I think MPI operations at each simulation step would be needed to e.g. reliably calculate a time series of the variance of a simulation variable within each PP cell because the average in each PP cell would have to be communicated at each simulation step so processes can calculate the variance of their simulation cells for that step.

Minor comments

The authors did not show initialization times for different test cases but would initialization of the PP system be feasible to perform e.g. every 10th simulation time step? The use case would be e.g. adaptive mesh refinement of simulation grid which would require the cell/process information to be updated between simulation and PP grids.

The authors, as well as the original Zängl et al. reference, use the term unstructured grid for the simulation grid. I would perhaps hesitate to call it unstructured because it seems that cells cannot be divided into an arbitrary number of smaller cells, for example. While the number of neighbors of a cell can vary due to mesh refinement the grid seems to be structured e.g. in the sense that every possible cell of the grid, i.e. its relative size, location and relative position to other cells, can be uniquely represented by a single integer. This was done for a cartesian grid in doi:10.1016/j.cpc.2012.12.017 where each cell can be divided into 2N cells along each edge (instead of N here), where N is a positive integer.

Technical suggestions/corrections

Page 4, line 16: und -> and

Page 6, figure caption: used for the calculation of averages. -> used for diagnostics in figures 2 and 3.

Answers to question at geoscientific-model-development.net/peer_review/review_criteria.htm

Does the paper address relevant scientific modelling questions within the scope of GMD? yes

Does the paper present a model, advances in modelling science, or a modelling protocol that is suitable for addressing relevant scientific questions within the scope of EGU? yes

Does the paper present novel concepts, ideas, tools, or data? yes

Does the paper represent a sufficiently substantial advance in modelling science? yes

Are the methods and assumptions valid and clearly outlined? yes

Are the results sufficient to support the interpretations and conclusions? yes

Is the description sufficiently complete and precise to allow their reproduction by fellow scientists (traceability of results)? with a few additions, yes

Do the authors give proper credit to related work and clearly indicate their own new/original contribution? yes

Does the title clearly reflect the contents of the paper? yes

Does the abstract provide a concise and complete summary? yes

Is the overall presentation well structured and clear? yes

Is the language fluent and precise? yes

Are mathematical formulae, symbols, abbreviations, and units correctly defined and used? yes

Should any parts of the paper (text, formulae, figures, tables) be clarified, reduced, combined, or eliminated? after traceability is improved, no

Are the number and quality of references appropriate? with SWMF reference, yes

Is the amount and quality of supplementary material appropriate? yes

---

## Referee Comment (RC2) · Anonymous Referee #2 · 12 Aug 2016

general comments: The authors describe an in-situ method of postprocessing and claim improved throughput and reduced disk space requirements when compared to current practices. But they do not appear to actually compare their results to the current practice. Does this new method in fact save time or disk space requirements? Are scientists willing to reduce the number of model fields written to disk if derived fields are computed in-situ?

---

## Author Comment (AC1) · 2 Sep 2016

We thank Ilja Honkonen (**Referee #1**) for his comments on the manuscript and for the insight provided by his point of view. The clarifications we include in the revised manuscript really help to improve it.

**Overall the paper is quite good but a some clarifications would allow the presented work to be compared with existing literature in space sciences where similar work already exist. My answers to GMD review criteria are at the end.**

**The authors implement a module for ICON modeling framework that allows run-time post-processing (PP), for which they use the term online diagnostics, of simulation results. Supported PP operations are the calculation of spatial average, spatial sum, discrete probability density function (PDF) and a two variable joint PDF. These can be calculated over a rectangular (in latitude and longitude) grid defined by the user. In a parallel run, if a cell of the PP grid spans more than one simulation cell, data required by PP operations is communicated between all processes that have simulation cells within the PP cell.**

**The term post-processing seems applicable here as the diagnostic module does not affect simulation results. The same term is also used e.g. in Open-FOAM documentation which shows an example of streamlines calculated at run-time (http://www.openfoam.com/version-v3.0+/post-processing.php retrieved on 2016-08-01). On the other hand the presented module could also be described as a one-way coupled model configurable with Fortran namelists.**

We are a bit reluctant to use the term post-processing here because:
1) We want to clearly distinguish between the "on-line post-processing" and classical post-processing of output data subsequent to a simulation, because in our community, post-processing is mostly associated with "off-line post-processing".
2) It is correct that there is no feedback from the on-line diagnostic tools to the model variables. But as the MESSy framework was introduced to make exactly this feedback possible (see also below), we want to stress that the MESSy submodels can be divided into *process submodels* influencing the results, the purely *diagnostic* ones, and the submodels providing the underlying infrastructure.
3) Moreover, ICON itself contains already a specific facility to apply post-processing to data just before the output. With our naming ("diagnostic") we want to make clear that the data produced by the *diagnostic* submodels is available in the memory (also for other submodels, see below) and not a concluding post-processing step.

**Overall it seems that the presented implementation can be divided into two distinct parts: 1) A new one-way coupled model for the MESSy framework that allows new variable(s) to be added to the framework relatively easily and allows calculations performed by the model to be chosen at the start of simulation. 2) A method for moving data from simulation grid to the PP module grid in a transparent way that supports different grid geometries, cell sizes and aggregation of all data from simulation cells into overlapping PP cells.**

What you refer to as item 1) is an already existing infrastructure submodel of MESSy, namely CHANNEL (Jöckel et al., 2010), which is used for memory and meta-data management, output control, and checkpointing. This submodel has been extended to meet the needs of the new basemodel ICON (as described in Section 2.3 and Appendix A1 of the manuscript).

Your item 2) is the novel *diagnostic* submodel GRAGG (GRid AGGregation), which utilises CHANNEL (cf. item 1) for meta-data management, memory access and output, i.e. for the definition of new variables (so called CHANNEL *objects*).

Indeed, we never thought of the GRAGG submodel as a transparent re-gridding tool, but as all submodels in the MESSy framework can access data in the memory via the CHANNEL submodel, it could be seen this way. Thank you for this insight. As there is already a generic MESSy submodel for re-gridding available, we should think of a further generalisation.

We include the idea of a more generalised re-gridding in Section 5 "*Ongoing developments*", which we rename to "*Ongoing and future developments*":

(p. 16, l. 6)

"*Last, but not least, as there is already a re-gridding tool available in the MESSy frame-work, an integration into the GRID submodel (Kerkweg and Jöckel, 2015) with further generalisation of the aggregation developed in this study seems beneficial.*"

**Major comments**

**Item 1) above looks like a regular MESSy component which uses the same grid as the basemodel, would that be a fair statement?**

The MESSy submodel CHANNEL incorporates the concept of *representations* (see Jöckel et al., 2010), which define the underlying geometric structure of *objects* (i.e. variables). Those can be defined as required. Thus, basemodel variables are naturally defined on the native grid geometry of the basemodel. Nevertheless, any MESSy submodel can define its own geometries.

**Can several PP modules run simultaneously on different sets of processes?**

MESSy is not a classical external coupler, although it contains also an infrastructure submodel for external coupling (Multi Model Driver, MMD, Kerkweg and Jöckel, 2012b). For the *diagnostic* submodels discussed here, several of those can be applied internally coupled, i.e., in sequence on the same task set.

**Can a PP module use data from other PP modules or only simulation results i.e. non-PP or basemodel data?**

[Figure]

CHANNEL *objects* can be accessed from *all* submodels, implying also that submodel A can access *objects* from submodel B.

**Is two-way coupling of PP modules possible, i.e. can two PP modules use each others' output data as their own input?**

Yes, because CHANNEL *objects* can be accessed from everywhere.

**Item 2) above allows a PP module to use a different grid from the one used by basemodel. In the presented work, cells of PP grid are larger than in simulation grid, is it possible to make PP grid cells smaller than in simulation grid?**

For technical reasons, this is currently not possible in GRAGG, because the intention of this submodel is data reduction.

**Taken together the functionality of items 1) and 2) are quite similar to e.g. the Space Weather Modeling Framework (SWMF, doi:10.1029/2005JA011126) which handles two-way data exchange between separate models running in parallel on different grids, although SWMF seems to lack the ability to collect all data from multiple smaller cells of the source grid into one cell of the destination grid. It would be helpful to clarify how the functionality presented here relates to that available in SWMF.**

We were not aware of the SWMF, which was initiated at about the same time as the MESSy framework, but we know some frameworks and couplers used in climate science. We think the framework and purpose of MESSy was exhaustively described in various articles (Jöckel et al. 2005, 2010, 2016). However, we add a short paragraph to our manuscript, which briefly relates MESSy to similar frameworks and couplers and

gives some examples of recent applications. References to comparable frameworks in climate science and the SWMF (See *References* below) are included in Section 2.2 of the revised manuscript.

Addressing the points above, we add the following paragraphs to the revised Section 2.2:

(p. 4, l. 9)

"*In geoscientific modelling, coupling of multi-institutional codes with generally different domain decompositions is a widely used approach for building model systems. In general, either* external *couplers or frameworks for* internal *coupling are applied. An extended classification of coupling methods can be found in Appendix A of Kerkweg and Jöckel (2012b) and in Jöckel (2012). An overview of different coupling techniques in Earth System Modelling is presented by Valcke et al. (2012). Common* external *couplers in the Earth System Model community are, e.g., OASIS3 (Valcke et al., 2006; Valcke, 2013), OASIS4 (Redler et al., 2010), and CPL6 (Craig et al., 2005), as used in the Community Climate System Model version 3 (CCSM3, Collins et al., 2006). Widely used examples for* internal *coupling are the Earth System Modeling Framework (ESMF, Collins et al., 2005) and the Community Climate Model version 4 (CCSM4, Gent et al. 2011). This approach is also used in space weather modelling with the Space Weather Modeling Framework (SWMF, Tóth et al., 2005). Recently, Hanke et al. (2016) developed the C-library YAC (Yet Another Coupler), which provides parallelised and efficient algorithms for grid transformation, interpolation, and data exchange.*

*In contrast to the coupling of "domains", MESSy was originally developed to work on the same spatial domain and parallel domain decomposition as the basemodel, applying a formalised process based operator splitting (Jöckel et al., 2005). The original imple-*

[Figure]

*mentation was intended to equip the atmospheric general circulation model ECHAM5 (Roeckner et al., 2006) with additional processes for atmospheric chemistry (EMAC, ECHAM5/MESSy Atmospheric Chemistry model, Jöckel et al., 2006, 2010). Operator splitting as* internal *coupling method is implemented (implicitly and less formalised) in the numerical model codes anyway, to integrate the different processes. However, the operator splitting approach of MESSy proves more powerful, also allowing for coupling of different domains, e.g., demonstrated by the integration of an ocean subsystem (Pozzer et al., 2011). An extension by Kerkweg an Jöckel (2012b) allows for one-way coupling of different spatially nested domains using a server-client approach with point-to-point communication. Extension into a two-way nested atmospheric chemistry model system is currently under development.* "

**The support, or whether there is any, for temporal analysis within the PP module was unclear to me. Is it possible to process simulation data over several time steps while calculating one diagnostic output, e.g. can the minimum and maximum values of a variable over a certain time range in each cell of the PP grid be determined? This would determine the min and max values more reliably than calculating them from saved model output which presumably would not be feasible to do for every simulation time step.**

In GRAGG, a temporal analysis is not included yet. However, the generic submodel CHANNEL already supports temporal operations (MIN, MAX, AVE, ...) over the output interval (see Section 2 and the Supplement of Jöckel et al. 2010). This facility can also be used for the variables calculated in GRAGG.

In Section 3 we add:

(p. 4, l. 9)

[Figure]

"*Note that a temporal aggregation is not envisaged for GRAGG, because the generic CHANNEL submodel already allows time aggregation operations, such as minimum, maximum, average and standard deviation, etc. over the output time interval (see Section 2 and the Supplement of Jöckel et al. 2010). This facility can also be used for the variables calculated in GRAGG.*"

**If the above is possible, then are MPI operations within each PP grid cell available at each simulation time step, or only at the end of a PP step? I think MPI operations at each simulation step would be needed to e.g. reliably calculate a time series of the variance of a simulation variable within each PP cell because the average in each PP cell would have to be communicated at each simulation step so processes can calculate the variance of their simulation cells for that step.**

The calculations of GRAGG and VISOP are currently triggered as requested by the user via namelist, either **every** time-step, or only every regular output time-step. For the calculation of correct time average, minimum, maximum, standard deviation, etc. w.r.t time, the trigger needs to be set to "every time-step". This provides the flexibility to reduce the expensive MPI operations to the minimum required. We plan to automate this in that way that an output request of temporal statistics (MIN, MAX, AVE, ...) via CHANNEL automatically triggers the corresponding calculations in every time-step.

**Minor comments**

**The authors did not show initialization times for different test cases but would initialization of the PP system be feasible to perform e.g. every 10th simulation time step? The use case would be e.g. adaptive mesh refinement of simulation grid which would require the cell/process information to be updated between simulation and PP grids.**

Since we currently do not use adaptive meshes, the weights for the grid transformations are calculated only once during the initialisation phase. An extension for time varying adaptive meshes is currently not planned.

**The authors, as well as the original Zängl et al. reference, use the term unstructured grid for the simulation grid. I would perhaps hesitate to call it unstructured because it seems that cells cannot be divided into an arbitrary number of smaller cells, for example. While the number of neighbors of a cell can vary due to mesh refinement the grid seems to be structured e.g. in the sense that every possible cell of the grid, i.e. its relative size, location and relative position to other cells, can be uniquely represented by a single integer. This was done for a cartesian grid in doi:10.1016/j.cpc.2012.12.017 where each cell can be divided into 2N cells along each edge (instead of N here), where N is a positive integer.**

The grid really seems structured, but the neighbour cells can not be deducted from the memory layout. As the grid is constructed from an icosahedron, resulting in vertices surrounded by five triangles and vertices surrounded by six triangles. The neighbourhood structure has to be explicitly stored, and hence the grid is indeed unstructured (https://en.wikipedia.org/wiki/Types_of_mesh).

**Technical suggestions/corrections**

**Page 4, line 16: und -> and**

Oh, the German tongue is breaking trough ;-) . Is corrected.

**Page 6, figure caption: used for the calculation of averages. -> used for diagnostics in figures 2 and 3.**

The referee is right that the grid is not only used for the calculation of averages, we update the sentence in the manuscript as follows:

*"Overlaid is the user-defined regular coarse grid (0.5 × 0.5) used to perform the on-line diagnostics in this study."*

*References*

[revised manuscript text omitted]

---

## Author Comment (AC2) · 2 Sep 2016

We thank the anonymous **Referee #2** for this concise, however, incisive review. The questions raised are very important. We answer them in the following, and adapt the manuscript to express more precisely our appraisal to these important topics.

**General comments**

**The authors describe an in-situ method of post-processing and claim improved**

[Figure]

**throughput and reduced disk space requirements when compared to current practices. But they do not appear to actually compare their results to the current practice.**

We did not intend to conduct any quantitative measurements w.r.t. throughput and disk space resulting from the application of on-line diagnostics. Numbers would depend on which and how many on-line diagnostics are applied. With the anticipation that data, which are "aggregated" in some way – may it be temporal or spatial average, column integrated values, any combination of variables, ... – have a smaller volume than the raw 3D model output, better throughput and lower disk space requirements are obvious. How this reduction in data volume is achieved, is another question. Currently, there are suggestions to use data compression (on-line or as post-processing) to decrease data volume and reduce the network and storage footprint (Kuhn et al., 2016; Baker et al., 2016).

An in-situ "post-processing" approach can reduce data volume. In practice, we anticipate a combination of the on-line "post-processing" and a limited set of "raw" data, which both could also undergo some on-line or off-line compression (lossy or lossless). As during the simulation checkpointing is anyway required, also a re-calculation of certain periods of the simulation with more output data could be a solution, when interesting features and evolution is detected in the in-situ "post-processing" data.

**Does this new method in fact save time or disk space requirements?**

If this approach saves disk space depends on which and how many on-line tools are applied, and which raw model fields are written out in addition. But when we can get rid of a large part of raw 3D output this is an opportunity to save huge expenses.

For the GRAGG submodel, as applied in this study, the aggregation has been applied

on a 0.5 × 0.5 target grid, using 10 × 10 bins for the joint PDF. In this specific case, the data volume of the raw output is about 4 times the output of the jPDF. In case of averaging variables onto the coarser grid, the data volume is reduced by a factor of of 187.

**Are scientists willing to reduce the number of model fields written to disk if derived fields are computed in-situ?**

We do not know if scientists are willing to do so, but latest with the advent of Exascale computer systems, scientist applying then-state-of-the-art ESMs may have no choice but to reduce the volume of output data in some way. Of course, this will largely change the overall workflow. Today, many simulations are conducted with data output, which is not necessary for the primary scientific question to be answered. However, it offers the opportunity for the scientist or another group to conduct further analyses on these additional data and maybe find some interesting insights.

We do not suggest to do all nowadays post-processing on-line, but we want scientists to think about, which data is needed, which analyses are useful, and which of them could be done already on-line. The tricky part is to find a balance between data reduction by reducing disk output through on-line analyses, and data which need to be written out in full resolution. With checkpointing, which is anyway required, scientists may have the opportunity to restart simulations of a certain period, when they find interesting features. Some additional data may be needed of course for quality checking.

The additional time to be invested for the development and testing of the on-line tools could potentially be saved from reduced efforts in post-processing and data handling (e.g., de-/archiving).

We are convinced that a combination of on-line tools, data compression and re-calculation from checkpoint data will shape the workflow in the next decades.

We reformulate and extend the *Introduction* of the manuscript:

(p. 1, l. 23)

[revised manuscript text omitted]